# Disparate Effects of Stressors on Met-Enkephalin System Parameters and on Plasma Concentrations of Corticosterone in Young Female Chickens

**DOI:** 10.3390/ani14152201

**Published:** 2024-07-29

**Authors:** Colin Guy Scanes, Krystyna Pierzchala-Koziec

**Affiliations:** 1Department of Biological Science, University of Wisconsin Milwaukee, Milwaukee, WI 53211, USA; 2Department of Animal Physiology and Endocrinology, University of Agriculture in Kraków, Mickiewicza 24/28, 30-059 Kraków, Poland

**Keywords:** Met-enkephalin, stress, PENK expression, chickens, corticosterone

## Abstract

**Simple Summary:**

There are multiple physiological changes in response to stress across vertebrate species. These shift metabolism and organ functioning such that the animals can withstand the stress. Ours results show that nutritional deprivation and crowding stimulate both the activities of the opioid system and the adrenocortical axis stress response in chickens.

**Abstract:**

The effects of stressors were examined on Met-enkephalin-related parameters and plasma concentrations of corticosterone in 14-week-old female chickens. Water deprivation for 24 h was accompanied by a tendency for increased plasma concentration of Met-enkephalin while plasma concentrations of corticosterone were elevated in water-deprived birds. Concentrations of Met-enkephalin were reduced in the anterior pituitary gland and adrenal gland in water-deprived pullets while proenkephalin (PENK) expression was increased in both tissues. There were changes in the plasma concentrations of Met-enkephalin and corticosterone in pullets subjected to either feed withholding or crowding. Concentrations of Met-enkephalin were increased in the anterior pituitary gland but decreased in adrenal glands in pullets subjected to crowding stress. The increase in the plasma concentrations of Met-enkephalin was ablated when the chickens were pretreated with naltrexone. However, naltrexone did not influence either basal or crowding on plasma concentrations of corticosterone. In vitro release of Met-enkephalin from the anterior pituitary or adrenal tissues was depressed in the presence of naltrexone. It was concluded that Met-enkephalin was part of the neuroendocrine response to stress in female chickens. It was concluded that stress influenced the release of both Met-enkephalin and corticosterone, but there was not complete parallelism.

## 1. Introduction

The hypothalamo-pituitary-adrenocortical (HPA) axis is usually considered to encompass the hypothalamus releasing corticotrophin releasing hormone (CRH) in response to stressors. This, in turn, stimulates the release of adrenocorticotrophic hormone (ACTH) from the anterior pituitary gland, and finally, the ACTH stimulates production of glucocorticoids (cortisol in many mammals and corticosterone in rodents and birds) from adrenocortical cells (reviewed in [1]).

There is a growing interest in using biomarkers for stress, depression, and fatigue, for instance, in people who have had strokes [2,3,4]. Examples of such biomarkers are elevated basal serum concentrations of the glucocorticoid, cortisol, together with pro-inflammatory cytokines, high-sensitivity C-reactive protein, ferritin, and neopterin [2]. While Met-enkephalin is not at present included in the list of putative biomarkers for stress in people who have had strokes [2,3], opioid peptides have been linked to stress-related human conditions including gastro-intestinal and inflammatory disorders [5]. Moreover, there is substantial evidence that Met-enkephalin is released in response to stressors such as parturition in cattle [6]. In addition, hypoxic stress increases Met-enkephalin release and synthesis in human neonatal chromaffin cells in vitro [7].

Met-enkephalin exerts an analgesic effect [8] but also has other effects. Met-enkephalin depressed hepatic catalase and glutathione peroxidase (Gpx) activities in male mice but increased these activities in females [9]. Met-enkephalin was previously called opioid growth factor due its ability to influence cell proliferation [10]. For instance, Met-enkephalin enhanced proliferation of splenic B-lymphocytes in male mice but inhibited it in female mice [11]. Met-enkephalin exerts a role in immune cell functioning with, for instance, effects on lymphocyte motility [12], the natural killer cellular concentration of cAMP [13], and T-lymphocyte proliferation [14]. Moreover, immune cells produce Met-enkephalin and, thereby, act as analgesics on peripheral sensory nerve terminals [15,16]. The immune role of Met-enkephalin is ancient, preceding the divergence of Protostomes and Deuterostomes 670 million years ago [17] with both Met-enkephalin and the delta opioid receptor present in the mollusc, the octopus [18].

Met-enkephalin is an opioid endogenous peptide and belongs to the enkephalins system. The synthesis of its precursor, proenkephalin (PENK), is under *PENK* (proenkephalin gene) expression. Biologically active Met-enkephalin exists in the tissues and blood as a native (free) form composed of five amino acids together with total PENK derived forms. Lightman and Young proposed that Met-enkephalin is part of the neuroendocrine system responding to stress [19]. There are interactions between the actions of hormones of glucocorticoids and those of Met-enkephalin [20].

To the best of our knowledge, there are no reports of the effects of stresses on plasma and tissue concentrations of Met-enkephalin in chickens or other birds, this being despite the importance of stress and animal well-being to poultry production. The present study examined the effects of a series of stresses (water deprivation, feed withdrawal, darkness, and crowding) on plasma, together with hypothalamic, anterior pituitary, and adrenal, concentrations of Met-enkephalin in pullets. A further objective was to examine the effects of the opioid receptor antagonist, naltrexone, to influence either Met-enkephalin or corticosterone concentrations. In addition, the effects of stressors and/or naltrexone on Met-enkephalin-related parameters and plasma concentrations of corticosterone were compared.

## 2. Materials and Methods 

All animal procedures were conducted with prior institutional ethical approval in accordance with the Local Institutional Animal Care and Use Committee (IACUC). The animal study protocol 120/2013 was approved by the Institutional Review Board and the First Local Ethical Committee on Animal Testing in Krakow, Poland.

### 2.1. Animals

The studies were performed on 14 weeks of age female chickens (line—ISA Brown), weighing 1.30 ± 0.10 kg. During the acclimatization period, the birds were housed in individual cages (60 × 60 × 40 cm) with feed (a commercial diet) and water available ad libitum. The chickens were maintained in a controlled environment with a temperature of 20 °C and a photoperiod of 12L:12D (lights on from 7.00 a.m. to 7.00 p.m.). 

### 2.2. Blood Sampling

Blood samples (2 mL) were obtained by venipuncture from the right branchial vein. The blood samples were divided into two sub-samples in polypropylene tubes containing either heparin (1000 IU) for corticosterone or EDTA 2.7 nmole mL^−1^ together with citric acid (17.7 nmole mL^−1^) and aprotinin (Trasylol, 200 KIU mL^−1^) for Met-enkephalin. After centrifugation (30 min at 4 °C and 4000× *g*), plasma was stored at −80 °C until further processing.

### 2.3. Tissue Sampling

Immediately after completion of the treatment periods, the birds were decapitated under anesthesia (injection of 20 mg kg b.w.^−1^ of ketamine plus 2 mg kg b.w.^−1^ of xylazine). The hypothalamus, pituitary gland, and adrenal glands were dissected. They were then weighed and divided into three sub-samples for the determination of Met-enkephalin (native and total forms) concentrations, *PENK* gene expression in some studies, and in vitro release of Met-enkephalin. Tissue fragments were homogenized in phosphate buffer, pH 6.5, and centrifuged (4000× *g*, 4 °C, 20 min). The supernatants were stored at −80 °C until further processing.

### 2.4. Experimental Design 

All studies (see below and Table 1) were approved by ILKE with the following protocol numbers (120/2013). 

Study 1. Effects of water deprivation. 

Study 1A. Effects on Met-enkephalin parameters together with plasma concentrations of corticosterone. Female chickens were controls or subjected to deprivation of water (for 24 h) (Table 1). 

Study 1B. Effects of water deprivation and naltrexone on release of Met-enkephalin in vitro from hypothalamic, anterior pituitary, and adrenal tissues either in control pullets or in birds after 24 h of water deprivation (as in study 1A).

Study 2. Effects of feed deprivation on Met-enkephalin parameters together with plasma concentrations of corticosterone. Female chickens were controls or deprived of feed (for 24 h) (Table 1).

Study 3. Effects of darkness (light deprivation) on Met-enkephalin parameters together with plasma concentrations of corticosterone. Female chickens were subjected to controls or subjected to darkness (deprived of light) for 24 h (Table 1). 

Study 4. Effect of crowding and/or naltrexone on Met-enkephalin parameters in pullets. There were four treatment groups: 1. controls that were individually caged pullets and receiving vehicle administration, 2. crowded chickens with five per cage for 30 min, 3. naltrexone (3 mg kg b.w.^−1^) injected i.v. and sampled after 30 min, and 4. birds receiving naltrexone and then subjected to crowding (Table 1). 

### 2.5. Hormone Assays

Concentrations of Met-enkephalin (native) and peptides containing Met-enkephalin motifs (total) in the tissues and plasma were determined by radioimmunoassay using the method of Pierzchała and Van Loon [21]. Met-enkephalin-containing peptides (total) were hydrolyzed with trypsin (1 mg mL^−1^, 37 °C, for 30 min), and then carboxypeptidase B (5 mg mL^−1^) plus trypsin inhibitor (2.5 mg mL^−1^) were added for 15 min. Native and total Met-enkephalins were separated on Porapak Q (100–120 mesh, Waters Corporation, Milford, CT, USA) in 2 mL of absolute ethanol, lyophilized, and assayed after reconstitution in 100 μL of 0.06 M phosphate buffer (pH 6.5). The assay entailed the addition of 50 μL Met-enkephalin antibody (rabbit, 1:10,000) and 50 μL of ^125^I-Met-enkephalin and incubation at 4 °C for 24 h. Then, 50 μL of antibody (rabbit γ-globulin) was added followed by incubation for 30 min at 4 °C. Separation of the bound from the unbound fractions was completed by the addition of 250 μL of 25% polyethylene glycol (PEG 8000), incubation for 30 min, and finally centrifugation (2000× *g*, 4 °C, 20 min). The supernatants were discarded, and the radioactivity of pellets was counted in a γ-counter (Wizard, Wallac Oy, Turku, Finland). 

The plasma concentration of corticosterone was determined by radioimmunoassay using a commercial kit (DRG Diagnostic, Marburg, Germany) with an intra-assay variation of 10.2% and an inter-assay variation of 7.1%.

### 2.6. Proenkephalin Gene (PENK) Expression

Prehybridization: Proenkephalin gene (*PENK*) expression was estimated by the modified method of Lightman and Young [19]. Frozen fragments of hypothalamus, pituitary, and adrenal tissues were sliced (14 μm sections) using a Leica cryostat microtome (Leica Biosystems Nussloch GmbH, Heidelberg, Germany) (−22 °C). The sections were thaw-mounted on gelatin-covered microscopic slides and stored for 3 days at −20 °C. Then, tissue sections were thawed and fixed in 4% formaldehyde in phosphate-buffered saline (PBS; pH 7.4) for 10 min. Sections were acylated for 10 min in triethanolamine/acetic anhydride (0.25%), dehydrated by immersion through graded ethanol (70%, 80%, 95%, and 100%), and then air dried.

After prehybridization, a synthetic deoxyoligonucleotide, complementary to the fragment of proenkephalin (PENK), was labeled using ^35^S-dATP (1200 Ci/nmol) to obtain a specific activity about 4 × 10^6^ cpm μL^−1^. Hybridization was performed during 20 h in a humidified chamber at 37 °C. Then, the sections were washed once in saline-sodium citrate buffer for 10 min, then four times for 15 min, each in SSC/50% formamide at 40 °C, rinsed in saline-sodium citrate buffer and distilled water at room temperature, and air-dried. The sections were exposed to Kodak film for four weeks (−80 °C). The photo-stimulated luminescence (PSL) density of the irradiated plates was measured with BAS-1000 readout system, Fujifilm, Tokyo, Japan). The PSL mm^−2^ at the resultant film images was determined using a computer image analysis system.

### 2.7. In Vitro Met-Enkephalin Release

Met-enkephalin secretion from fragments of tissues was estimated with the published method [21] with some modifications. Briefly, fragments of tissues (20–30 mg) sliced by microtome were placed into plates with 1 mL of Krebs–Ringer bicarbonate buffer as a medium. After a 20 min preincubation period, tissues were incubated at 37 °C for four successive 20 min periods in 500 μL medium according to the sequence: 1. basal medium, 2. stimulating medium with 100 nM of naltrexone, 3. basal medium, and 4. medium with potassium. The concentrations of Met-enkephalin in the basal media were not significantly different, and so the results were pooled and presented as Met-enkephalin release under basal conditions.

### 2.8. Statistical Analysis

The results are presented as means ± SEM. The data from study 1A and study 3 were analyzed by Student’s t-test. Differences between treatments in study 1B and study 4 were analyzed by two-way ANOVA with means separated by Tukey’s honest significance test. Differences between times in study 2 were analyzed by repeated measures ANOVA with means separated by Tukey’s honest significance test or by paired *t*-tests.

## 3. Results 

### 3.1. Effects of Water Deprivation on Plasma Concentrations of Met-Enkephalin (Native and Total) and Corticosterone Together with Tissue Concentrations of Met-Enkephalin and PENK Expression

There was no effect of water deprivation on plasma concentrations of either Met-enkephalin or peptides containing Met-enkephalin motifs in female chickens (Table 2). The plasma concentrations of corticosterone were increased (*p* < 0.01) by 116% (Table 2). 

Concentrations of Met-enkephalin were decreased (*p* < 0.001) (by 51.1%) in the anterior pituitary gland in water-deprived female chickens (Table 2). Similarly, adrenal concentrations of Met-enkephalin were depressed (by 28.8%) (*p* < 0.05) in water-deprived female chickens (Table 2). In contrast, *PENK* expression was increased (*p* < 0.001) in the anterior pituitary gland and the adrenal gland of water-deprived chickens (Table 2). There were no effects of water deprivation on either the concentrations of Met-enkephalin or *PENK* expression in the hypothalamus (Table 2). 

### 3.2. Effects of Water Deprivation In Vivo in Pullets in the Presence or Absence of Naltrexone In Vitro on Release of Met-Enkephalin In Vitro

There was considerably greater release of Met-enkephalin from hypothalamic and adrenal tissue than from anterior pituitary tissue by 22.5- and 17.3-fold, respectively (Figure 1). The release of Met-enkephalin from hypothalamic tissue from water-deprived chickens was lower (*p* < 0.01) by 40.6% than in control chickens (Figure 1). Similarly, the release of Met-enkephalin was lower (*p* < 0.05) from anterior pituitary and adrenal tissue from birds that were water deprived. Moreover, the release of Met-enkephalin from anterior pituitary and adrenal tissue was depressed (*p* < 0.05) when incubated in the presence of naltrexone in vitro by 30.9 and 34.3%, respectively (Figure 1). 

It was possible that the in vitro release of Met-enkephalin (Figure 1) was artifactual, reflecting the effect of water deprivation on tissue concentrations of Met-enkephalin (Table 3). To preclude this possibility, in vitro release is expressed as a percentage of the tissue content (Table 3). There was greater (*p* < 0.05) release of Met-enkephalin in vitro from the adrenal than the hypothalamic and, in turn, than the anterior pituitary tissues when the in vitro release was expressed as a percentage of the tissue content (Table 3). The in vitro release from the three tissues was lower (*p* < 0.05) from birds from which water was withheld (Table 3). 

### 3.3. Effects of Feed Deprivation on Plasma Concentrations of Met-Enkephalin and Corticosterone

Plasma concentrations of native Met-enkephalin were decreased (by 43%) (*p* < 0.01), and total Met-enkephalin was not changed (a tendency to increase being observed) in feed-deprived female chickens (Table 4). Plasma concentrations of corticosterone were increased (by 134%) (*p* < 0.001) (Table 4). However, in birds resupplied with feed for 2 h, plasma concentrations of native Met-enkephalin returned to the control value, while those of corticosterone remained increased (Table 4).

### 3.4. Effects of Light Deprivation (Darkness) on Plasma and Tissue Concentrations of Met-Enkephalin Together with PENK Expression and Plasma Concentrations of Corticosterone

Plasma concentrations of native Met-enkephalin, total peptides containing Met-enkephalin motifs, and corticosterone were increased in pullets exposed to darkness for 24 h by similar percentages, 24.5%, 21.2%, and 22.4%, respectively (Table 5). Similarly, concentrations of Met-enkephalin, peptides containing Met-enkephalin motifs, and *PENK* expression were increased in darkness-exposed pullets in the anterior pituitary gland by 81.1%, 90.3%, and 130%, respectively (Table 5). In contrast, there were decreases in the hypothalamic and adrenal concentrations of Met-enkephalin (by 52.3% and 54.5%) and peptides containing Met-enkephalin motifs (by 50.1% and 47.4%) and adrenal *PENK* expression (by 39.0%) (Table 5).

### 3.5. Effect of Crowding on Plasma and Tissue Concentrations of Met-Enkephalin and Plasma Concentrations of Corticosterone in Pullets

Plasma concentrations of native Met-enkephalin were increased (*p* < 0.01) in chickens subject to crowding (by 38.5%) (Table 6). In the presence of naltrexone, there were lower (*p* < 0.001) plasma concentrations of Met-enkephalin in birds that received naltrexone and were subjected to crowding than those subjected to crowding alone (Table 6). There were shifts in tissue concentrations of Met-enkephalin in chickens subjected to crowding with decreases (*p* < 0.001) in the hypothalamus (by 59.3%) and adrenal glands (54.2%) and increases (*p* < 0.01) in the anterior pituitary gland (75.5%) (Table 6). There were no effects of naltrexone on tissue concentrations of Met-enkephalin in control birds (Table 6). However, there were no increases in plasma concentrations of Met-enkephalin in chickens subjected to crowding and pretreated with naltrexone (Table 6). Similarly, there were no decreases in concentrations of Met-enkephalin in the hypothalamus or increases in the anterior pituitary gland with chickens subjected to crowding but pretreated with naltrexone (Table 6). 

There were increases (by 79.1%) (*p* < 0.01) in the plasma concentrations of corticosterone in chickens subject to crowding (Table 6). There were no effects of prior naltrexone administration on plasma concentrations of corticosterone in either controls or chickens subjected to crowding for 30 min (Table 6). 

## 4. Discussion

Not only is Met-enkephalin present in multiple organs in mammals, including the brain and spinal cord [22], heart, kidney epithelium, intestinal epithelium, intestinal smooth muscle cells, skeletal muscle [23], spleen [24], intermediate lobe of the pituitary gland [22], adrenal medulla [21,25], and lungs [21], but Met-enkephalin is also released from at least the following: heart, intestine, and skeletal muscle [23]. Moreover, *PENK* expression is reported in multiple tissues including the periventricular nucleus in the hypothalamus [26], luteal cells [27], astrocytes [28], skin cells [29], ovarian follicular cells [27], and mesodermal tissues during organogenesis [30]. Met-enkephalin is present in both the chromaffin cells and the nerve terminals of the splanchnic nerve [31]. Similarly, there is wide distribution of *PENK* expression in chickens [32,33]. The source of Met-enkephalin in plasma is not clear, there being no effects of adrenal demedullation [25].

Figure 2 shows the structure of chicken pro-enkephalin [33]. In addition, the figure shows a series of dibasic amino-acids residues, presumptive sites for peptidase/convertase actions to generate peptides. Sites that generate Met-enkephalin sequences and a single Leu-enkephalin sequence are shown. Possible peptides containing Met-enkephalin motifs that would be generated by convertase are also shown, namely, YGGFMRF and YGGFMRSF. These are conserved across the tetrapods, indicating their biological importance [33]. It is recognized that there are other biologically active peptides containing Met-enkephalin motifs that may be generated from pro-enkephalin [34]. What is not clear is the biological activities of YGGFMRF and YGGFMRSF and any other peptides containing Met-enkephalin motifs.

Changes in circulating concentrations of Met-enkephalin, such as those reported in the present studies, may reflect the following (see Figure 3): Changed synthesis of pro-enkephalin at levels of expression and/or translation;Changed generation of Met-enkephalin and other PENK-derived peptides by peptidases;Changed release of Met-enkephalin and other PENK-derived peptides;Shifts in peptidases either generating or degrading Met-enkephalin and other PENK-derived peptide in the circulation and/or at target tissues.

These putative control sites are consistent with data from the present studies. For instance, *PENK* expression was increased in both the anterior pituitary and adrenal glands of water-deprived (Table 2) and restrained chickens [35]. Moreover, concentrations of Met-enkephalin in the anterior pituitary and adrenal glands were decreased with water deprivation in contrast to the increased *PENK* expression in the same tissues (Table 2).

Met-enkephalin has been proposed as part of the neuroendocrine system responding to stress [19]. The plasma concentrations of Met-enkephalin are increased while tissue concentrations of Met-enkephalin are shifted in mammals in response to stressors such as the administration of hypertonic saline in rats [19], isolation stress in lambs [36], nicotine administration in rats [21,24], opioid withdrawal in rats [19], and restraint stress in rats [37]. However, with the exception effect of restraint [35], to the best of our knowledge, there are no reports on the effects of stressors on Met-enkephalin in chickens or other avian species. In the present studies, decreases in the plasma concentrations of Met-enkephalin were observed only with pullets subjected to feed deprivation (Table 4). Met-enkephalin concentrations in the hypothalamus, anterior pituitary gland, and adrenal gland were affected by stresses as follows: decreased in the anterior pituitary gland and adrenal gland following water deprivation (Table 2), decreased in the hypothalamus and adrenal gland following crowding (Table 6), and increased in the anterior pituitary gland following darkness (Table 5) or crowding (Table 6) stress. In water-deprived pullets, there were increases in *PENK* expression in the anterior pituitary gland and adrenal gland (Table 2). Moreover, there was decreased in vitro release of Met-enkephalin from hypothalamic tissue from chickens subjected to water deprivation (Figure 1, Table 3). 

In vivo, naltrexone blocked the effects of crowding stress on concentrations of Met-enkephalin in the plasma, hypothalamus, anterior pituitary gland, and adrenal gland (Table 6). There were in vitro effects of naltrexone on the release of Met-enkephalin from hypothalamic, anterior pituitary and adrenal explants partially reversing the lower rates of Met-enkephalin release in tissues from water-deprived chickens (Figure 1). This was consistent with Met-enkephalin being auto-inhibitory.

As would be expected, the plasma concentrations of corticosterone in chickens were increased by stressors. There is strong evidence that stresses are followed by increased plasma concentrations of corticosterone predominantly from studies in young meat-type chickens [1]. In the present studies, the stress-induced increase in plasma concentrations of corticosterone occurred irrespective of whether the stressors were water deprivation (Table 2), fasting (Table 4), or crowding (Table 6) together with restraint stress [35]. Fasting has previously been demonstrated to evoke increases in plasma concentrations of corticosterone in chickens [38,39]. Similarly, in chickens subjected to protein deprivation, there were increases in corticosterone synthesis [40]. Moreover, crowding has been reported to increase plasma concentrations of both corticosterone and cortisol /in laying hens [41]. 

Table 7 summarizes the directionality of effects of stressors on plasma concentrations of Met-enkephalin, plasma concentrations of corticosterone, tissue concentrations of Met-enkephalin, and *PENK* expression in the hypothalamus, anterior pituitary glands, and adrenal glands. Plasma concentrations of corticosterone were consistently increased by each stressor. Broadly, plasma concentrations of Met-enkephalin were elevated by each stressor (except feed deprivation). In a congruent manner, both the plasma and anterior pituitary concentrations of Met-enkephalin were elevated by stressors. There was some relationship between the plasma concentrations of Met-enkephalin and those of corticosterone in females (R^2^ = 0.443, *p* < 0.05) but none in males (R^2^ = −0.055, *p* = 0.650). 

In summary, a series of stressors influenced Met-enkephalin synthesis, processing [enzymatic hydrolysis of PENK or proopiomelanocortin (POMC) in pituitary], and plasma concentrations of Met-enkephalin in pullets. This was despite the presumed short half-life of native Met-enkephalin in the circulation and tissues, this being less than 2 min based on studies in rhesus monkeys [43]. The changes of Met-enkephalin concentrations were seen even after 24 h stress. It is suggested that this opioid is responsible for the attenuation of severe stress responses including high blood pressure, cardiovascular changes, pain, metabolic, and immune processes. It is suggested that this opioid is an indicator (biomarker) of stress responses based on the stress-related shifts in Met-enkephalin parameters in female chickens.

## Figures and Tables

**Figure 1 animals-14-02201-f001:**
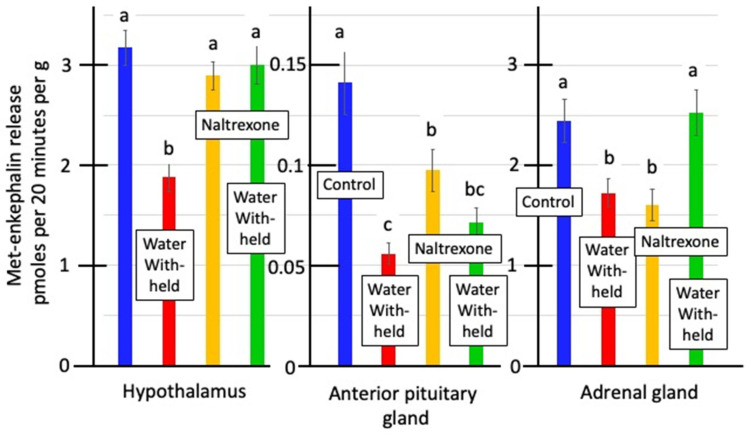
Effect of water deprivation in vivo and/or in vitro treatment with naltrexone on in vitro release of native Met-enkephalin from chicken hypothalamic, anterior pituitary, and adrenal explants. [Blue—control; red—water withheld; yellow—naltrexone supplementation; green—naltrexone + water withheld; a, b, c—different letters indicate difference (*p* < 0.05)].

**Figure 2 animals-14-02201-f002:**
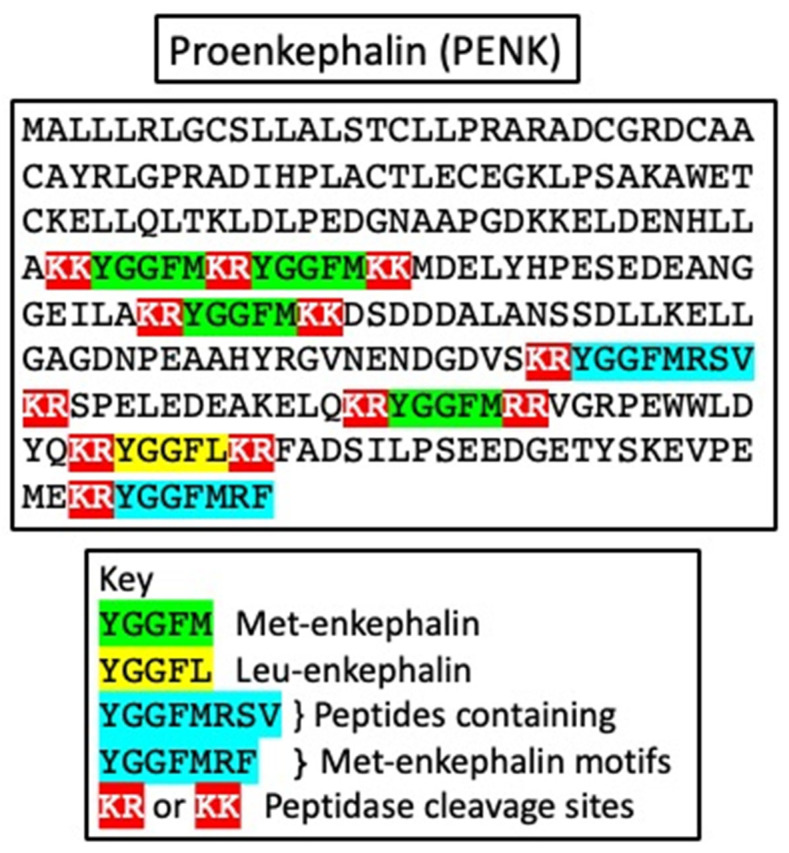
Structure of proenkephalin (adapted from deduced structure from Genbank accession number XM_040664746.2) showing sites from which Met- and Leu-enkephalin are generated by peptidase action as bibasic amino-acid residue sites together with sites from which peptides containing Met-enkephalin motifs can be generated.

**Figure 3 animals-14-02201-f003:**
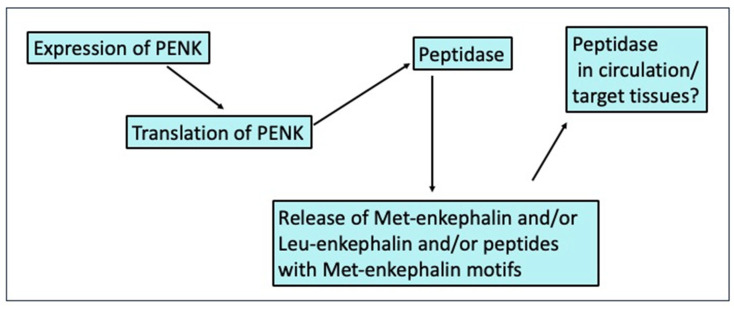
Control sites for the availability of Met-enkephalin, Leu-enkephalin, and peptides containing Met-enkephalin motifs.

**Table 1 animals-14-02201-t001:** Deprivation treatments and Met-enkephalin parameters determined in 14-week-old female chickens.

Study	Duration	*n*	Treatments Groups	Plasma Concentrations	Tissue Concentrations of Met-Enkephalin	Tissue PENK Expression
1. Water deprivation	24 h	6	1. Control 2. Water withheld	Corticosterone Met-enkephalin peptides containing Met-enkephalin motifs	Hypothalamus, anterior pituitary gland, adrenal glands ^A^	Hypothalamus, anterior pituitary gland, adrenal glands
2. Feed deprivation	24 h	6	1. Control 2. Feed deprivation	Corticosterone Met-enkephalin peptides containing Met-enkephalin motifs	-	-
3. Light deprivation (darkness)	24 h	5	1.Control 2. Light deprivation	Corticosterone Met-enkephalin peptides containing Met-enkephalin motifs	Hypothalamus, anterior pituitary gland, adrenal glands	Hypothalamus, anterior pituitary gland, adrenal glands
4. Space deprivation (crowding)	30 min	5	1. Control 2. Crowding 3. Naltrexone 4. Crowding and naltrexone	Corticosterone Met-enkephalin	Hypothalamus, anterior pituitary gland, adrenal glands	Hypothalamus, anterior pituitary gland, adrenal glands

^A^ In addition, in vitro release of Met-enkephalin was determined in the presence or absence of naltrexone; see Study 1B.

**Table 2 animals-14-02201-t002:** Effects of in vivo water deprivation for 24 h on plasma and tissue concentrations of Met-enkephalin and *PENK* expression [mean ± (*n* = 6) SEM] in 14-week-old female chickens.

Tissue and Parameter	Control	Water Withheld (24 h)
Plasma Concentrations		
Native Met-enkephalin (pmoles L^−1^)	50.2 ± 7.9	68.7 ± 6.2
Total peptides containing Met-enkephalin motifs (pmoles Met-enkephalin equivalents L^−1^)	705 ± 69	570 ± 49
Corticosterone (nmoles L^−1^)	11.1 ± 2.0	38.5 ± 2.6 ***
Tissue concentration of Native Met-enkephalin (pmoles g^−1^)		
Hypothalamus	371 ± 29	285 ± 31
Anterior pituitary gland	822 ± 63	402 ± 29 ***
Adrenal gland	95 ± 10	68 ± 6 *
*PENK* expression as % of control		
Hypothalamus	100 ± 1.0	102 ± 1.0
Anterior pituitary gland	100 ± 2.1	530 ± 38 ***
Adrenal gland	100 ± 2.0	328 ± 9 ***

Difference between water withheld and control chickens, * *p* < 0.05, *** *p* < 0.001.

**Table 3 animals-14-02201-t003:** Effects of in vivo water deprivation for 24 h and/or in vitro naltrexone treatment on in vitro release of Met-enkephalin expressed as a % of tissue content per 20 min [mean ± (*n* = 5) SEM] in 14-week-old female chickens.

Tissue	In Vivo Treatment	Relative Met-Enkephalin Release (% of Tissue Content per 20 min)	
		In vitro treatment	
		Control	Naltrexone
Hypothalamus as a % of tissue content per 20 min			
	No treatment (control)	0.86 ± 0.05 ^b^	0.78 ± 0.04 ^b^
	Water withheld	0.66 ± 0.05 ^a^	1.05 ± 0.05 ^c^
Anterior pituitary gland as a % of tissue content per 20 min			
	No treatment (control)	0.017 ± 0.002 ^b^	0.012 ± 0.001 ^a^
	Water withheld	0.014 ± 0.001 ^a^	0.18 ± 0.002 ^b^
Adrenal gland as a % of tissue content per 20 min			
	No treatment (control)	2.56 ± 0.23 ^b^	1.69 ± 0.16 ^a^
	Water withheld	2.54 ± 0.20 ^b^	3.72 ± 0.34 ^c^

^a,b,c^ Different superscript letters indicate difference *p* < 0.05.

**Table 4 animals-14-02201-t004:** Effects of feed deprivation on plasma concentrations of Met-enkephalin [mean ± (*n* = 6) SEM] and corticosterone [mean ± (*n* = 6) SEM] in 14-week-old layer-type young female chickens.

	Control	Feed Deprivation	Re-Feed for 2 h
Native plasma concentrations of Met-enkephalin (pmoles L^−1^)	56.0 ± 5.7 ^b^	32 ± 4.3 ^a^	63.0 ± 7.7 ^b^
Total plasma concentration of peptides containing Met-enkephalin motifs (pmoles Met-enkephalin equivalents L^−1^)	660 ± 49	720 ± 49	750 ± 59
Plasma concentrations of corticosterone (nmoles L^−1^)	16.0 ± 1.3 ^a^	37.4 ± 4.3 ^b^	34.8 ± 3.9 ^b^

^a,b^ Different superscript letters indicate difference *p* < 0.05.

**Table 5 animals-14-02201-t005:** Effects of darkness (light deprivation) on plasma and tissue concentrations of Met-enkephalin together with tissue PENK expression and plasma concentrations of corticosterone [mean ± (*n* = 5) SEM] in pullets.

	Control	Dark Stress
Plasma Concentrations of Mean ± SEM (*n* = 5)		
Native Met-enkephalin (pmole L^−1^)	31.0 ± 0.88	38.6 ± 0.90 ***
Total peptides containing Met-enkephalin motifs (pmole L^−1^)	462 ± 10.7	560 ± 15.5 ***
Corticosterone	17.4 ± 0.51	21.3 ± 0.80 **
Native tissue concentrations of Met-enkephalin (pmole g^−1^) mean ± SEM (*n* = 5)		
Hypothalamus	411 ± 4.01	196 ± 2.05 ***
Anterior pituitary gland	826 ± 4.91	1496 ± 35 **
Adrenal gland	112 ± 4.14	55.8 ± 0.86 ***
Tissue concentrations of total peptides containing Met-enkephalin motifs (pmole g^−1^) mean ± SEM (*n* = 5)		
Hypothalamus	3673 ± 71.5	1673 ± 41.5 **
Anterior pituitary gland	7623 ± 110	14,507 ± 223 ***
Adrenal gland	726 ± 21.5	382 ± 8.5 ***
*PENK* expression (as % of controls) (*n* = 3)		
Hypothalamus	100 ± 6.9	97.1 ± 3.5
Anterior pituitary gland	100 ± 3.2	230 ± 9.5 ***
Adrenal gland	100 ± 5.6	61.0 ± 5.6 **

** *p* < 0.01, *** *p* < 0.001.

**Table 6 animals-14-02201-t006:** Effects of space deprivation (crowding) in the presence or absence of naltrexone (3 mg/kg b.w.) pretreatment on plasma and tissue concentrations of Met-enkephalin together with plasma concentrations of corticosterone [mean ± (*n* = 5) SEM] in young female chickens.

	Control	Crowding for 30 min	Control	Crowding
	Sham		Naltrexone pretreatment	
Plasma concentrations				
Native Met-enkephalin (pmoles L^−1^)	37.7 ± 8.8 ^a,b^	52.2 ± 6.3 ^b^	45.9 ± 7.3 ^b^	30.9 ± 5.0 ^a^
Corticosterone (nmoles L^−1^)	13.4 ± 1.8 ^a^	24.0 ± 4.4 ^b^	16.7 ± 1.8 ^a^	24.7 ± 3.7 ^b^
Tissue concentrations of native Met-enkephalin (pmoles g^−1^)				
Hypothalamus	403 ± 44 ^b^	164 ± 26 ^a^	359 ± 45.7 ^b^	342 ± 53.6 ^b^
Anterior pituitary gland	930 ± 139 ^a,b^	1632 ± 261 ^b^	1044 ± 148 ^a,b^	600 ± 73 ^a^
Adrenal gland	118 ± 13 ^c^	54 ± 7 ^a^	108 ± 16 ^b,c^	73.1 ± 12 ^a,b^
*PENK* expression as % of control				
Hypothalamus	100 ± 4.8 ^a^	47.0 ± 2.3 ^c^	54.3 ± 2.8 ^b^	49.7 ± 1.9 ^b,c^
Anterior pituitary gland	100 ± 7.9 ^b^	244 ± 9.6 ^c^	105 ± 6.7 ^b^	39.6 ± 3.3 ^a^
Adrenal gland	100 ± 3.8 ^a^	62.5 ± 3.8 ^b^	114 ± 7.3 ^a^	66.7 ± 4.8 ^b^

^a,b,c^ Different superscript letters indicate differences, *p* < 0.05.

**Table 7 animals-14-02201-t007:** Effects of stressors on plasma concentrations of Met-enkephalin, plasma concentrations of corticosterone, tissue concentrations of Met-enkephalin, and *PENK* expression in female chickens.

Deprivation	Plasma Concentration of Met-Enkephalin	Plasma Concentration of Corticosterone	Tissue Concentration of Met-Enkephalin	*PENK* Expression
Females				
Hypothalamus				
Water ^T^	↑?	↑↑	→	→
Food ^U^	↑↑	↑↑	NA	NA
Darkness ^V^	↑	↑	↓↓	→
Crowding ^W^	↑?	↑	↓↓	↓↓
Anterior pituitary gland				
Water ^T^	↑?	↑↑	↑↑	↑↑↑
Darkness ^V^	↑	↑	↑↑	↑↑
Crowding ^W^	↑?	↑	↑↑	↑↑
Restraint ^X^	↑↑	↑↑	NA	↑↑
Morphine ^Y^	↓↓	↑↑	→	↓↓
Adrenal gland				
Water ^T^	↑?	↑↑	↓↓	↑↑
Darkness ^V^	↑	↑	↓↓	↓↓
Crowding ^W^	↑?	↑	↓↓	↓↓
Restraint ^X^	↑↑	↑↑	NA	↑↑
Morphine ^Y^	↓↓	↑↑	→	↓↓

^T^ From Table 2, ^U^ from Table 4 ^V^ from Table 5, ^W^ from Table 6, ^X^ [35] and ^Y^ [42] NA not available. ↑? possible increase, ↑ increased, ↑↑ large increase, ↑↑↑ very large increase, → no change, ↓↓ large decrease.

## Data Availability

Data are contained within the article.

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
