# Peer review of "Disparate Effects of Stressors on Met-Enkephalin System Parameters and on Plasma Concentrations of Corticosterone in Young Female Chickens"

_animals, 2024, doi:10.3390/ani14152201_

Round 1

Reviewer 1 Report

Comments and Suggestions for Authors

The experimental design and presentation of results are entirely satisfactory. However, the introduction gives no indication as to the role of enkephalins in stress responses and the discussion and conclusion give no suggestion that the results anything to what we already know from corticosteroids.

The introduction begins with a simple single sentence outlining the role of the HPA axis in the stress response. You then state that enkephalins are 'part of the endocrine response to stress' but give no indication as to their possible role. You must say something about the known actions of enkephalins ('opiod growth factors') in relation to (e.g.) nociception and immune function. This might give some indication as to what we might learn from enkephalins that we have not learnt from corticosteroids.

Author Response

Overall statement: We appreciate the reviewer’s helpful critique of our manuscript. To improve the clarity of the manuscript, two new tables have been added. The first summarizes the studies performed. The second new table present data on release of Met-enkephalin in vitro expressed as a percentage of the tissue concentration to preclude possible artifacts due to effects of water deprivation on tissue contents of Met-enkephalin.

Comment 1

The experimental design and presentation of results are entirely satisfactory.

Response: We thank the reviewer for their kind statement.

Comment 2

However, the introduction gives no indication as to the role of enkephalins in stress responses and the discussion and conclusion give no suggestion that the results add anything to what we already know from corticosteroids. 

Response:

We considered it important to include corticosterone to allow comparison with the data on Met-enkephalin.

Comment 3

The introduction begins with a simple single sentence outlining the role of the HPA axis in the stress response. You then state that enkephalins are 'part of the endocrine response to stress' but give no indication as to their possible role. You must say something about the known actions of enkephalins ('opioid growth factors') in relation to (e.g.) nociception and immune function. This might give some indication as to what we might learn from enkephalins that we have not

learnt from corticosteroids.

Response:

We thank the reviewer for the comment. There is now an extensive discussion of the actions of Met-enkephalin in the Introduction section including both nociception and immune functioning.

Reviewer 2 Report

Comments and Suggestions for Authors

Birds and pullets were used to meassure some stress parameters.

In general: The study is insufficiently described and the study cannot be repeated. There are no repetitions. 

Introduction: The objective is described insufficiently and there are ethical objections therefore. 

M&M: There are males, females, controls and test animals??? The numbers of the animals are missing. The housing is not described. The statistical analyses can be discussed. It is recommended to ask a statiticien for advice. 

Results: the text at the figures is insufficient to understand the figure.

Discussion: The conclusion need to be revised.

Author Response

Overall statement: We appreciate the reviewer’s helpful critique of our manuscript.

Comment 1

Birds and pullets were used to measure some stress parameters.

In general: The study is insufficiently described and the study cannot be repeated. There are no repetitions. 

Response:

The Materials and Methods section has been thoroughly revised to improve clarity A new table has been added; this summarizes the studies performed.

Comment 2

Introduction: The objective is described insufficiently and there are ethical objections therefore. 

Response

We respectively disagree with the reviewer.  According to the three RRR the number of chickens were sufficient but not excessive.  The aim of the study was to follow the activity of Met-enkephalin system in parallel with corticosterone under physiological and stressful conditions. If someone wants to repeat the study –there is no problem as the methods employed are clear.

Comment 3

M&M: There are males, females, controls and test animals??? The numbers of the animals are missing. The housing is not described. The statistical analyses can be discussed. It is recommended to ask a statistician for advice. 

Response We appreciate reviewer 2’s concerns. The materials and methods section has been expanded and clarified.  Particular attention has been focused on the description of the studies, the housing of the birds and the statistics section.

Comment 4

Results: the text at the figures is insufficient to understand the figure.

Response: We again appreciate reviewer 2’s concerns.  The legend for Figure 1 is expanded with additions including defining the colors employed.

Comment 5

Discussion: The conclusion needs to be revised.

Response

The conclusion has been revised.

Round 2

Reviewer 1 Report

Comments and Suggestions for Authors

This is a great improvement on the previous manuscript as it now explains how met-enkephalins can be involved in certain stress responses. I recommend that it is nearly acceptable for publication, although I am still concerned that, after the big build-up in the introduction, the discussion and conclusions are largely unchanged and still do not convince me that the recruiting  met-enkephalins as yet another stress monitor adds anything significant tp our understanding of the stress response.

Author Response

We thank the reviewer for their helpful comments.

This is a great improvement on the previous manuscript as it now explains how met-enkephalins can be involved in certain stress responses. I recommend that it is nearly acceptable for publication, although I am still concerned that, after the big build-up in the introduction, the discussion and conclusions are largely unchanged and still do not convince me that the recruiting  met-enkephalins as yet another stress monitor adds anything significant tp our understanding of the stress response.

Response: We have revised the discussion to improve its clarity.

Reviewer 2 Report

Comments and Suggestions for Authors

The paper is improved. Editing of the English text in the introduction is required.

Introduction

L56-57: If a difference between male and female in rats might excist, it is necessary to use female chickens in the whole text and title (L4,30,76,401).

L43,47 Is people a human being? 

Comments on the Quality of English Language

The paper is improved. Editing of the English text in the introduction is required

Author Response

We thank the reviewer for their helpful comments.

L56-57: If a difference between male and female in rats might exist, it is necessary to use female chickens in the whole text and title (L4,30,76,401).

Response: We have made all the requested changes.

L43,47 Is people a human being? 

Response: Yes

The paper is improved. Editing of the English text in the introduction is required

Response: We have carefully revised the introduction to improve the use of English.